# Cultural factors in mid- and later life volunteerism in the United States: A scoping review protocol

Patrick Ho Lam Lai[ID]1*, Briana White-Saul1, Chin-Yi Su[ID]2, Qiuchang (Katy) Cao3

1 Anne and Henry Zarrow School of Social Work, University of Oklahoma, Norman, Oklahoma, United States of America, 2 School of Social Work, Boston College, Chestnut Hill, Massachusetts, United States of America, 3 College of Social Work, Florida State University, Tallahassee, Florida, United States of America

* patricklai@ou.edu, patricklairesearch@gmail.com

## Abstract

### Introduction

Volunteerism in midlife and later life is linked to improved well-being for individuals and stronger communities, but participation rates differ widely across groups. Most existing research has focused on socioeconomic factors to explain volunteering, while cultural factors have received less attention and are often narrowly defined, usually in terms of religiosity. As a result, it remains unclear how culture is conceptualized and examined in relation to volunteering in later life. To address this gap, it is important to systematically review how cultural factors have been defined and studied in relation to volunteerism among older adults.

### Methods and analysis

This protocol describes a scoping review that will follow the JBI Scoping Review Framework and the PRISMA-ScR guidelines. The review aims to identify and map how cultural factors are conceptualized and operationalized in studies of formal and informal volunteerism among adults aged 50 and older in the United States. Searches will include several academic databases. Two reviewers will independently screen studies and extract data. Findings will be synthesized using descriptive and thematic analysis. Cultural factors identified across studies will be grouped into themes based on conceptual similarity to summarize how culture has been conceptualized and operationalized in U.S.-based volunteerism research and study. As this is a protocol, no results are reported.

**Registration:** This protocol has been registered on the Open Science Framework (OSF) (https://doi.org/10.17605/OSF.IO/4SZVU).

**Data availability statement:** No datasets were generated for this protocol. After study completion, review materials (including search strategies, screening decisions, data charting form, and extracted data) will be shared on OSF, subject to copyright and licensing restrictions.

**Funding:** The author(s) received no specific funding for this work.

**Competing interests:** The authors have declared that no competing interests exist.

## Introduction

Volunteerism during midlife and later life represents a significant form of social and civic participation, with clear benefits for both individuals and communities. Participation in volunteer activities is associated with improved mental health, reduced social isolation, and an enhanced sense of purpose among older adults [1–2]. At the community level, older volunteers make substantial contributions by strengthening social networks and community cohesion, enhancing public health, and addressing neighborhood challenges [1–2]. At the individual level, identifying the factors that influence volunteer participation in later life is essential for promoting healthy and productive aging, defined as meaningful participation in social and civic activities in later life that benefit both individuals and society [1,3,4].

Although the benefits of volunteering are well documented, participation in midlife and later life varies substantially across individuals and social contexts. Prior research has often emphasized socioeconomic resources, such as education and income, as key determinants of volunteer participation. For instance, individuals with higher levels of education, income, and social capital tend to have greater access to resources, time, and opportunities that facilitate participation in volunteering than those with lower socioeconomic status [2,5,6]. However, emerging evidence suggests that these resources do not fully explain observed variations in engaging in formal/ organized volunteering (e.g., providing service through RSVP, senior companions, foster grandparents programs) or informal/unorganized volunteering (e.g., helping a neighbor, self-initiated neighborhood clean-ups/projects) [2]. For example, studies using nationally representative data show that differences in volunteering persist even after accounting for socioeconomic factors, suggesting that additional factors, including cultural values, norms, expectations regarding volunteering, and religious influences, may shape volunteer participation in midlife and later life [2,7–9]. These patterns highlight the need to examine how culture is studied in relation to volunteerism in later life.

Despite growing recognition of culture as a relevant influence, its role in volunteer participation during mid and later life remains unclear in the literature. Cultural factors are operationalized inconsistently, often relying on single-item measures, and frequently lacking explicit theoretical grounding or employing very different theoretical frameworks, such as cultural capital [10] or cultural system [11]. Some studies operationalize cultural influences primarily through indicators of religious participation or religiosity [7,12,13]. While religion and spirituality are important aspects of cultural life, this approach overlooks other culturally embedded influences, such as values, norms, social expectations, identity, and the meanings attributed to volunteerism. Additional culture-related dimensions, including country or region of origin, settlement context, and immigration timing or generational status, are also pertinent to understanding volunteerism in midlife and later life. Across social science disciplines, culture has been conceptualized not as a fixed or monolithic attribute, but as a set of meanings and practices that are learned, interpreted, and enacted within specific social and historical contexts [14–15]. Adopting this flexible and inclusive understanding allows the review to capture various ways cultural factors have been

conceptualized and operationalized across populations and research contexts. Because cultural contexts are often unspecified in the existing literature on mid- and later-life volunteer participation, we aim to explore how cultural factors have been defined and measured in U.S. volunteerism research and scholarship. For the purpose of this review, screening will use a broad working interpretation of cultural factors. Studies will be considered potentially eligible when they explicitly refer to culture or related cultural constructs in relation to volunteerism. During data extraction, we will record the specific concepts, measures, examples, descriptions, and elaborations that each source presents as cultural. This approach supports consistency in study selection while remaining open to the diverse ways culture has been defined and operationalized in the literature.

This review focuses on the U.S. to ensure conceptual coherence and practical relevance. In the U.S., the midlife and older adult population is highly diverse in race, ethnicity, and immigration history, making it a critical context for understanding how culture is conceptualized and operationalized in volunteerism research. The purpose of this review is not to provide a global synthesis, but to map how cultural factors are conceptualized and operationalized within a single national context. Restricting the review to the U.S. also helps reduce cross-national variation in policy context, institutional structures, and meanings of volunteerism.

To reflect this diversity, the review examines both formal and informal forms of volunteerism. Organized volunteering is not always considered a culturally meaningful participation, particularly among U.S. populations from diverse cultural and different racial and ethnic backgrounds [2,16]. Instead, some individuals may express helping and civic contributions through informal volunteering, shaped by cultural norms such as familial obligation and community responsibility [17]. Including both informal and formal volunteering allows us to capture the diversity of cultural factors in existing studies across cultural contexts in the U.S.

Although volunteering promotes health and well-being among people of all ages, it may be especially important in mid- and later-life, particularly in reducing loneliness among widowed older adults [18–19]. From a developmental and life-course perspective, midlife is a critical period for generativity, during which individuals seek meaningful roles that contribute to others and to society through productive and civic activities, including volunteering [20]. Participation in volunteering during midlife may support psychosocial well-being by reinforcing purpose, social connection, and identity, while also shaping patterns of participation that carry forward into later life. As individuals transition into later adulthood, volunteering may take on additional significance as people experience changes in work roles, family responsibilities, health status, and social networks, including retirement, bereavement, and caregiving transitions. In these later stages, volunteering can help buffer psychosocial challenges by enhancing social connectedness and community embeddedness [21]. Exploring cultural factors that shape volunteering across midlife and older adults is particularly helpful in informing volunteer programming, recruitment, and retention strategies that support volunteer participation.

A preliminary search conducted in November 2025 using databases and platforms such as PubMed, Web of Science, Google Scholar, JBI Evidence Synthesis, and the Open Science Framework (OSF) did not identify any existing scoping reviews that systematically examine cultural factors influencing formal and informal volunteerism among adults aged 50 and older. Although numerous reviews have examined volunteering in mid- and later life, these reviews have primarily focused on outcomes, correlates, and benefits of volunteering in mid- and later life, including its physical and mental health benefits and quality of life [22–25], cognitive functioning [26], socio-ecological systems [27], and active aging [28]. Other reviews have focused on social, economic, and other factors contributing to volunteer engagement [29–30], related forms of cultural engagement (e.g., museum attendance [31]), or on volunteerism in non-U.S. contexts [28]. However, existing reviews do not synthesize how cultural factors are conceptualized and operationalized in volunteerism research. As a result, the current review literature provides limited guidance on how cultural factors are represented across studies of volunteer participation in mid- and later life in the U.S.

Accordingly, the objective of this scoping review is to identify and synthesize cultural factors examined in the literature on formal and informal volunteerism among adults aged 50 and older and to assess how these factors have been

conceptualized and operationalized in volunteerism research and literature in the U.S. The findings are intended to inform future research, practice, and policy efforts aimed at developing culturally responsive approaches to supporting volunteerism.

## Methods and analysis

### Overall study design

This study will employ a scoping review design to map the literature on cultural factors influencing volunteerism among adults aged 50 and older in the U.S. Scoping review methodology is appropriate given the conceptual and methodological heterogeneity of the existing literature and the exploratory nature of the research questions. The review will follow the JBI Scoping Review Framework [32] and will be reported in accordance with the PRISMA-ScR guideline.

The PRISMA-ScR guideline provides a structured approach for transparent reporting of scoping reviews, including clear documentation of eligibility criteria, search strategies, study selection, and data synthesis. Its checklist was used in the development of the study protocol (S1 Checklist) [33]. Following the JBI Scoping Review Framework, the review process will include six stages: (1) identifying the research question, (2) identifying relevant studies, (3) study selection, (4) charting the data, (5) collating, summarizing, and reporting the results, and (6) expert consultation. The purpose of this review is to map how cultural factors have been conceptualized and operationalized in volunteerism studies and research, rather than to evaluate study quality or estimate effect sizes.

### Study status and timeline

At the time of protocol submission, the review is in the planning and preparation stage. Search strategies have been developed and piloted, but formal title/abstract screening, full-text screening, and data charting have not been completed. We anticipate completing study selection and data charting by March 2026, with final synthesis and manuscript completion expected by May 2026.

**Step 1: Identifying the research question.** The research questions guiding this scoping review are:

1. What cultural factors have been defined, conceptualized, and operationalized in studies of formal and informal volunteerism among adults aged 50 and older in the U.S.?

2. Where are cultural factors incorporated and discussed in studies of volunteerism in midlife and later life (for example, within study design, measurement, analysis, or interpretation)?

These research questions were developed using the Population-Concept-Context (PCC) framework recommended by JBI. The population (P) is adults aged 50 and older, as this age range captures mid- and later-life stages when key life transitions commonly occur, and volunteer participation becomes particularly relevant. The concept (C) is cultural factors discussed. The context (C) includes formal volunteerism conducted through organizations and informal volunteerism involving unpaid community-based help. This framework was chosen to ensure the review captures how cultural factors are examined across diverse forms of volunteerism in midlife and later life within the U.S.

Together, these research questions are designed to summarize how cultural factors are currently represented in the empirical volunteerism literature, rather than to assess the strength, direction, or causal nature of their associations with volunteer participation. This approach is consistent with the exploratory aims of a scoping review and supports the identification of conceptual definitions, operationalized measures, as well as gaps in existing research.

**Step 2: Identifying relevant studies.** A comprehensive search strategy will be developed in consultation with a subject social work librarian. To improve feasibility and reduce overlap across sources, searches will be conducted in a focused set of core academic databases most relevant to the topic, including PubMed, PsycINFO, AgeLine, CINAHL, Social Services Abstracts, Scopus, and Web of Science. Grey literature sources will also be searched,

including dissertations and theses (e.g., ProQuest Dissertations & Theses Global), policy documents, government and organizational reports, technical reports, conference proceedings, and working papers. Backward and forward citation tracking will also be conducted for included studies to identify additional relevant sources.

This scoping review will include peer-reviewed empirical studies (e.g., quantitative, qualitative, and mixed-methods designs) and grey literature (e.g., reports, policy documents, dissertations). This approach allows the review to capture a comprehensive range of conceptualizations and operationalizations of cultural factors in volunteerism research, particularly where cultural discussions may not be formally tested in empirical designs.

Search terms will combine concepts related to volunteerism, aging, midlife, and later life, and cultural factors. The search strategy used as a pilot in PubMed is shown in Table 1. Search terms will be refined iteratively and reviewed by a subject librarian. The final search strings will be applied consistently across databases. Backward and forward citation tracking will be conducted for the final included studies to identify additional relevant literature that may not be captured through database searches alone. The initial search is planned for March 2026 and will be updated before the final analysis to incorporate the most recently published studies.

**Step 3: Study selection.** Data will be extracted and organized using Covidence. The titles and abstracts will be reviewed by two reviewers to determine if they meet the eligibility criteria listed below, with duplicate articles and excluded articles being omitted through Zotero. Once data compilation and the initial screening for eligibility have been completed, two reviewers will independently review the full texts of the articles to confirm that the eligibility criteria are met for each source. Reviewers will document reasons for exclusion during full-text screening to maintain transparency and consistency. Disagreements will be further discussed and resolved through organized team meetings.

*Eligibility criteria.* Studies of various designs and methodologies, including grey literature, will be considered for review. Because we focus on volunteerism in the U.S. context, only English-language studies with full-text access available through the University of Oklahoma, University of Oklahoma Health Campus, Florida State University, and Boston College library systems, or through publicly accessible sources such as Google Scholar and general Google search. If the full-text article is not accessible to the reviewers, it will be excluded.

The contents of the articles must follow the eligibility criteria presented in Table 2. Included studies must be explicitly examining culture, and they are related to formal and/or informal volunteering. There will not be a cap on the year that the articles were published to be eligible, as it is believed by the researchers that the value of research contributions on this subject does not expire. Searches will be updated prior to final analysis to identify newly published studies. Complete search strategies for all databases will be documented and reported in the final review.

*Screening reliability:* Inter-rater reliability will be checked after the pilot screening to assess how consistently reviewers apply the criteria. An agreement level of at least 70% will be considered acceptable, in line with prior scoping review practice [34]. If agreement is lower, the team will review a subset of disagreements to clarify the screening guidelines before moving forward. Final inclusion decisions will be made through discussion and consensus.

**Table 1. PubMed search strategy.**

| |
|---|
| *Search Terms* |
| (volunteer*[tiab] OR "informal helping"[tiab] OR "unpaid help"[tiab] OR "Volunteers"[Mesh]) AND (elder*[tiab] OR "older adult*"[tiab] OR senior*[tiab] OR aging[tiab] OR aged[tiab] OR "later life"[tiab] OR midlife[tiab] OR "middle aged"[tiab] OR "Aged"[Mesh]) AND (cultur*[tiab] OR acculturat*[tiab] OR "Culture"[Mesh] OR "Acculturation"[Mesh]) |
| *Search Limits* |
| English language |

**Table 2. Eligibility criteria.**

| Inclusion criteria | Exclusion criteria |
|---|---|
| Population: Studies examining adults aged 50 years or older, or populations explicitly described as mid-life or older adults, including studies with a clearly identifiable midlife or older adult subsample. | Population: Studies focusing exclusively on younger populations, or studies that do not specify participant age and describe the population as midlife or older adults. |
| Concept: Studies that explicitly address cultural factors (e.g., culture, cultural capital, acculturation, cultural values). | Concept: Studies that do not explicitly discuss cultural factors. |
| Context: Studies examining formal and/or informal volunteering or helping behaviors. | Context: Studies that do not examine volunteering or helping behaviors. |
| Study design: All study designs, including quantitative, qualitative, mixed-methods, and conceptual or theoretical works. | Study design: Not applicable. |
| Language: Studies published in English. | Language: Studies published in languages other than English. |
| Publication type: Empirical journal articles and grey literature (e.g., reports, policy documents, dissertations, conference proceedings). | Publication type: Studies will be excluded if the full text is not accessible. |
| Date of publication: No upper limit on publication year; searches conducted through now. | Date of publication: Not applicable. |
| Study location: Include studies in which the study population, or a clearly identifiable subsample, is located in the U.S. | Study location: Exclude studies in which the study population is located entirely outside the U.S., and no U.S.-based subsample can be clearly identified. |

**Step 4: Charting the data.** Data will be extracted using a standardized charting form developed, piloted, and refined by the research team before the full extraction. The included studies will be divided between two reviewers, and each reviewer will check the accuracy of the other's extracted data. Disagreements will be discussed and resolved by consensus at team meetings. Extracted information will include publication details, source type, study characteristics, cultural conceptualization and operationalization, theoretical frameworks, volunteer participation, and where cultural factors are discussed in each source (e.g., research questions, conceptual framework, measurement, analysis, interpretation, program description, or recommendations). For volunteer participation, we will capture status (e.g., participation vs non-participation), frequency or intensity, duration, type or domain (formal or informal), and, when available, willingness, motivation, or intention to volunteer. Cultural factors (e.g., values, norms, identity, religion or spirituality, language, acculturation, immigration-related factors, and culturally shaped meanings of helping or obligation) will be recorded as reported in the original source, with no new data generated. Explicit definitions or descriptions of cultural concepts will be recorded verbatim. The data charting domains are summarized in Table 3.

Descriptive statistics and thematic analysis will be used to identify and organize cultural factors examined across studies and to summarize how these factors are linked to volunteer participation.

**Step 5: Collating, summarizing, and reporting the results.** Extracted data will be synthesized using descriptive statistics and thematic analysis [35]. Descriptive statistics (e.g., frequencies, percentages, and ranges) will be used to summarize the types and prevalence of cultural factors examined across studies, allowing identification of both under-studied and well-studied areas. In addition, we report key characteristics of studies, including study design (e.g., qualitative, quantitative, or mixed-method studies); year range of publications; sample sizes; etc.

Further, we will conduct a thematic analysis [35] to organize and synthesize how cultural factors are conceptualized and operationalized in the literature. To explore the meaning, context, and interpretation of culture in the existing literature, we will first immerse ourselves in the extracted data, then generate codes and group them. After comparing codes to one another and to the extracted data, we will group codes into broader themes based on conceptual similarity (e.g., ikigai

**Table 3. Draft data extraction domains.**

| Domains | Extracted information |
|---|---|
| Publication details | author, year, source type |
| Study characteristics | study design, sample |
| Volunteer participation | status, frequency, duration, type |
| Cultural conceptualization | definition or description of culture |
| Cultural operationalization | measures, proxies, qualitative themes |
| Theoretical framework | theories referenced |
| Section in article | where cultural factors appear (design, research questions, measures, analysis, interpretation, program description, or recommendations) |
| Reported linkage | How culture relates to volunteering |

and filial piety). We will then define the themes to reflect patterns in how and where cultural factors are discussed across studies of volunteering in midlife and later life.

**Step 6: Expert consultation.** We will consult with both methodological and content experts to ensure the rigor and practical relevance of our study. We consult with a health or social sciences librarian on search strategies and key terms. Further, when data extraction is complete, we consult with experts in the field (e.g., leaders in aging services and volunteer programs) to support the interpretation of findings and inform the development of a framework on culture and volunteering.

## Consultation with stakeholders

Patients and the public were not involved in the design of this scoping review protocol. The review focuses on synthesizing existing literature.

## Ethics and dissemination

Ethics approval is not required, as this study involves analysis of published literature only. Findings will be disseminated through peer-reviewed publications and conference presentations.

## Discussion

This scoping review will provide a structured overview of how cultural factors have been conceptualized and operationalized across empirical studies and grey literature in U.S. volunteerism scholarship. By summarizing conceptualization and operationalization, the review will identify gaps and heterogeneity in how culture is addressed in the existing literature and establish a foundation for advancing culturally informed volunteerism research and program design in the U.S., particularly by highlighting cultural impacts that have been emphasized, underexamined, or inconsistently represented.

This review will identify dominant cultural constructs to support the refinement of cultural concepts and the development of more robust measurement strategies for future U.S.-based quantitative and qualitative research, as well as data collection. This is particularly relevant in the U.S., where midlife and older adult populations are increasingly diverse in race, ethnicity, immigration history, and cultural background, all of which shape meanings and expectations of social and civic participation.

Beyond research implications, the findings will inform practice and policy efforts aimed at promoting volunteer participation in later life in the U.S. community organizations, aging service providers, and volunteer programs may use insights from this review to design recruitment, retention, and engagement strategies that better reflect culturally diverse

motivations, meanings, and forms of helping. At the policy level, the review will provide evidence to guide the development of initiatives that support productive aging and strengthen volunteer infrastructures in ways that recognize cultural diversity across U.S. communities.

Key limitations include restricting inclusion to English-language sources and not conducting a critical appraisal, consistent with scoping review objectives.

## Registration

This protocol is registered with the OSF (https://doi.org/10.17605/OSF.IO/4SZVU).

## Supporting information

**S1 Checklist. Preferred reporting items for systematic reviews and meta-analyses extension for Scoping Reviews (PRISMA-ScR) checklist.**
(DOCX)

## Acknowledgments

Financial support was provided by the University of Oklahoma Libraries' Open Access Fund.

## Author contributions

**Conceptualization:** Patrick Ho Lam Lai, Briana White-Saul, Qiuchang (Katy) Cao.

**Investigation:** Patrick Ho Lam Lai, Briana White-Saul.

**Methodology:** Patrick Ho Lam Lai, Briana White-Saul, Chin-Yi Su.

**Project administration:** Patrick Ho Lam Lai.

**Supervision:** Patrick Ho Lam Lai.

**Writing – original draft:** Patrick Ho Lam Lai, Briana White-Saul, Chin-Yi Su, Qiuchang (Katy) Cao.

**Writing – review & editing:** Patrick Ho Lam Lai, Briana White-Saul, Chin-Yi Su, Qiuchang (Katy) Cao.

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
