## [Decision Letter · Decision Letter 0]

19 Mar 2026

PONE-D-26-06935Cultural Factors in Mid- and Later Life Volunteerism in the United States: A Scoping Review ProtocolPLOS One

Dear Dr. Lai,

Thank you for submitting your manuscript to PLOS ONE. After careful consideration, we feel that it has merit but does not fully meet PLOS ONE’s publication criteria as it currently stands. Therefore, we invite you to submit a revised version of the manuscript that addresses the points raised during the review process.

We look forward to receiving your revised manuscript.

Kind regards,

Olushayo Oluseun Olu

Academic Editor

PLOS One

Reviewers' comments:

Reviewer's Responses to Questions

**Comments to the Author**

1. Does the manuscript provide a valid rationale for the proposed study, with clearly identified and justified research questions?

Reviewer #1: Yes

2. Is the protocol technically sound and planned in a manner that will lead to a meaningful outcome and allow testing the stated hypotheses?

Reviewer #1: Yes

3. Is the methodology feasible and described in sufficient detail to allow the work to be replicable?

Reviewer #1: Yes

4. Have the authors described where all data underlying the findings will be made available when the study is complete?

Reviewer #1: Yes

5. Is the manuscript presented in an intelligible fashion and written in standard English?

Reviewer #1: Yes

6. Review Comments to the Author

You may also provide optional suggestions and comments to authors that they might find helpful in planning their study.

Reviewer #1: The protocol addresses an important and timely topic and is generally well written, clearly structured, and methodologically sound. The authors provide a clear rationale for examining how cultural factors are conceptualized and operationalized in volunteerism research among midlife and older adults. The use of established methodological guidance, including the JBI Scoping Review Framework and PRISMA-ScR reporting standards, is appropriate and strengthens the transparency and reproducibility of the proposed review. The research questions are clearly stated, the eligibility criteria are well described, and the proposed procedures for screening, data extraction, and synthesis are logically presented. Overall, the protocol demonstrates careful planning and has the potential to make a valuable contribution to the literature on volunteerism and aging. The comments below are offered to further improve clarity and methodological transparency.

Introduction

1. Clarification of the definition of “cultural factors” (lines 68–80)

The manuscript discusses culture broadly, including values, norms, identity, religion, and immigration-related factors. While this broad framing is appropriate, it would be helpful to clarify how “cultural factors” will be interpreted during screening and data extraction. Given that the review aims to map how culture is conceptualized in the literature, briefly outlining the domains or types of factors that will be considered cultural may help improve consistency in study selection.

2. Justification for restricting the review to the United States (lines 84–87)

The protocol limits the review to studies conducted in the United States and briefly notes that this is due to the diversity of the U.S. midlife and older adult population. While this is reasonable, the rationale could be expanded slightly. Clarifying that the objective is to map conceptualizations of culture within a single national context, rather than conduct a global synthesis, would strengthen the justification for this restriction.

Methods and Analysis

3. Number of databases proposed for the search (lines 165–171)

The protocol proposes searching 14 academic databases. While comprehensive coverage is important, several of these databases index overlapping journals. The authors may wish to consider focusing on a smaller set of core databases most relevant to the topic, supplemented by grey literature and citation tracking. This would likely maintain adequate coverage while improving feasibility. Similarly, as the search strategies will likely need to be reported for each database in the supplementary materials, the large number of proposed databases may result in extensive and repetitive supplementary files. Streamlining the database list may also help simplify the reporting and improve clarity.

7. PLOS authors have the option to publish the peer review history of their article (what does this mean?). If published, this will include your full peer review and any attached files.

Reviewer #1: **Yes:** Robert Lubajo

---

## [Author Response · Author response to Decision Letter 1]

24 Mar 2026

Reviewer #1

Comment 1. Clarification of the definition of “cultural factors” (lines 68–80)

The manuscript discusses culture broadly, including values, norms, identity, religion, and immigration-related factors. While this broad framing is appropriate, it would be helpful to clarify how “cultural factors” will be interpreted during screening and data extraction. Given that the review aims to map how culture is conceptualized in the literature, briefly outlining the domains or types of factors that will be considered cultural may help improve consistency in study selection.

Our Response:

Thank you for this helpful suggestion. We clarified in the manuscript that screening will use a broad working interpretation of cultural factors and that data extraction will record the specific concepts, measures, examples, and descriptions that each source presents as cultural. We also added brief examples to further clarify the types of cultural factors that may be captured (Lines 83–89 and 232–233).

Comment 2. Justification for restricting the review to the United States (lines 84–87)

The protocol limits the review to studies conducted in the United States and briefly notes that this is due to the diversity of the U.S. midlife and older adult population. While this is reasonable, the rationale could be expanded slightly. Clarifying that the objective is to map conceptualizations of culture within a single national context, rather than conduct a global synthesis, would strengthen the justification for this restriction.

Our Response:

We appreciate this suggestion. Hence, we expanded the rationale for focusing on the United States by clarifying that the review aims to map how culture is conceptualized and operationalized within a single national context, rather than to conduct a global synthesis (Lines 93–96).

Comment 3. Number of databases proposed for the search (lines 165–171)

The protocol proposes searching 14 academic databases. While comprehensive coverage is important, several of these databases index overlapping journals. The authors may wish to consider focusing on a smaller set of core databases most relevant to the topic, supplemented by grey literature and citation tracking. This would likely maintain adequate coverage while improving feasibility. Similarly, as the search strategies will likely need to be reported for each database in the supplementary materials, the large number of proposed databases may result in extensive and repetitive supplementary files. Streamlining the database list may also help simplify the reporting and improve clarity.

Our Response:

We reviewed the database list and streamlined the search to focus on core databases most relevant to the topic, while retaining grey literature searching and backward and forward citation tracking. This revision improves feasibility and clarity while maintaining broad coverage (Lines 175–181). Thank you!

---

## [Editor Report · Decision Letter 1]

6 Apr 2026

Cultural factors in mid- and later life volunteerism in the United States: A scoping review protocol

PONE-D-26-06935R1

Dear Dr. Lai,

We’re pleased to inform you that your manuscript has been judged scientifically suitable for publication and will be formally accepted for publication once it meets all outstanding technical requirements.

Kind regards,

Olushayo Oluseun Olu

Academic Editor

PLOS One
---

## [Editor Report · Acceptance letter]

PONE-D-26-06935R1

PLOS One

Dear Dr. Lai,

I'm pleased to inform you that your manuscript has been deemed suitable for publication in PLOS One. Congratulations! Your manuscript is now being handed over to our production team.

Kind regards,

on behalf of

Dr. Olushayo Oluseun Olu

Academic Editor

PLOS One